# Selection Criteria for Pressurized Intraperitoneal Aerosol Chemotherapy (PIPAC) Treatment in Patients with Peritoneal Metastases

**DOI:** 10.3390/cancers14102557

**Published:** 2022-05-23

**Authors:** Aurélie Balmer, Daniel Clerc, Laura Toussaint, Olivia Sgarbura, Abdelkader Taïbi, Martin Hübner, Hugo Teixeira Farinha

**Affiliations:** 1Department of Visceral Surgery, Faculty of Biology and Medicine UNIL, Lausanne University Hospital (CHUV), Rue du Bugnon 46, 1011 Lausanne, Switzerland; aurelie.balmer@unil.ch (A.B.); daniel.clerc@chuv.ch (D.C.); laura.toussaint@chuv.ch (L.T.); martin.hubner@chuv.ch (M.H.); 2Surgical Oncology Department, Montpellier Cancer Institute (ICM), University of Montpellier, F-34298 Montpellier, France; olivia.sgarbura@icm.unicancer.fr; 3Institut de Recherche en Cancérologie de Montpellier (IRCM), INSERM U1194, Université de Montpellier, F-34298 Montpellier, France; 4Digestive Surgery Department, Dupuytren Limoges University Hospital, CNRS, XLIM, UMR 7252, F-87000 Limoges, France; abdelkader.taibi@hotmail.fr

**Keywords:** PIPAC, peritoneal cancer, carcinomatosis, peritoneal surface malignancies, peritoneal metastases

## Abstract

**Simple Summary:**

Standard treatment protocol for PIPAC consists of three procedures and completion of treatment has been shown to be prognostic of improved survival. The aim of this study was to identify predictors for completion of treatment. This retrospective multicentric cohort study included all patients with peritoneal metastases undergoing PIPAC at three PIPAC expert centers. Overall, 183 patients had 517 PIPACs. Bimodal treatment was found as an independent predictive factor for completing the three procedures (OR = 4.202, 95%CI [1.813, 10.630], *p* < 0.001), as well as prior bowel obstruction (OR = 0.389, 95%CI [0.153, 0.920], *p* = 0.037). In conclusion, the absence of ascites and prior bowel obstruction can help to select patients suitable for PIPAC. Best results seem to be achieved when PIPAC is combined with systemic chemotherapy.

**Abstract:**

Background: The standard treatment protocol for PIPAC consists of three procedures. Completion of treatment has been shown to be prognostic of improved survival. The aim of this study was to identify predictors for completion of treatment. Methods: Retrospective multicentric cohort study of patients with peritoneal metastases undergoing PIPAC in three PIPAC expert centers. Per protocol (PP) treatment was defined as patients receiving ≥3 PIPACs and was compared to patients receiving <3. Results: Overall, 183 patients had 517 PIPACs. The main reasons for stopping PIPAC were disease progression in 50% patients, bowel obstruction in 15%, patient’s refusal to pursue in 10%, conversion to cytoreductive surgery in 7%, and medical reasons in 8%. Overall, 95 patients (52%) had PP treatment. The PP median OS was 17 vs. 7 months, *p* = 0.001. PP patients had r ascites (410 ± 100 mL vs. 960 ± 188 mL, *p* = 0.001), no prior history of bowel obstruction (12% vs. 24%, *p* = 0.028), and more bimodal treatment (39% vs. 13%, *p* < 0.001). After multiple regression, bimodal treatment was found as an independent predictive factor for completing PP (OR = 4.202, 95%CI [1.813, 10.630], *p* < 0.001), along with prior bowel obstruction (OR = 0.389, 95%CI [0.153, 0.920], *p* = 0.037). Conclusion: The absence of ascites and prior bowel obstruction can help to select patients suitable for PIPAC. Best results seem to be achieved when PIPAC is combined with systemic chemotherapy.

## 1. Introduction

Peritoneal metastasis (PM) remains a therapeutic challenge with ominous prognosis mainly due to poor response to systemic chemotherapy [1,2,3]. Intraperitoneal chemotherapy delivery has been proposed as an alternative therapeutic option to enhance drug concentrations in tissue and to reduce systemic toxicity [4].

Pressurized intraperitoneal aerosol chemotherapy (PIPAC) was introduced in 2011 as an innovative intraperitoneal drug delivery method in several experimental and clinical studies [5,6,7]. The actual standard treatment consists of three PIPAC procedures planned every 4–6 weeks in association with systemic chemotherapy and can be pursued depending on tolerance and treatment response [8,9]. Recent prospective and retrospective cohort studies show good tolerance of repeated PIPAC treatment and the rare occurrence of intraoperative and postoperative complications [6,10,11,12].

Preliminary oncological results show encouraging tumoral response to PIPAC after the completion of three applications [13,14]. However, many patients do not complete the full treatment course and, for various reasons, have to stop after only one or two procedures, hence receiving only limited benefit [15]. We aimed to investigate predictive factors of PIPAC treatment discontinuation in order to better select patients who are most likely to benefit from a complete PIPAC treatment course.

## 2. Materials and Methods

This is a retrospective multicentric cohort study including consecutive patients undergoing PIPAC treatment from January 2015 to January 2020 in Lausanne University Hospital (CHUV), Switzerland; Dupuytren Limoges University Hospital, France; and Montpellier Cancer Institute (ICM), France. Patients with peritoneal metatases (PM) form various origins (ovarian, colorectal, gastric, hepato-pancreato-biliary (HPB), and mesothelioma) were included and all indications for PIPAC treatment were decided in a multidisciplinary tumor board in line with current practice consensus [6]. The exclusion criteria were age <18 years old and patient’s refusal to participate.

Per protocol (PP) treatment was defined as patients receiving ≥3 PIPACs and was compared to patients with 1 or 2 PIPACs (<3 PIPAC group).

### 2.1. Outcomes

Reasons to stop PIPAC were stratified into 10 groups: intraperitoneal progression, extraperitoneal progression, bowel obstruction, patient’s refusal to pursue, conversion to curative cytoreductive surgery, non-access, absence of disease, PIPAC complication, death, or other medical reasons. Other medical reason included, for example, infections, pulmonary embolism, or myocardial infarction.

Baseline variables (demographics, previous systemic IV chemotherapy, symptoms before PIPAC, surgical details, and postoperative complications according to Clavien-Dindo [16]) were compared between the two groups to investigate factors which could predict discontinuation of treatment.

### 2.2. Data Management

Demographic, surgical, and oncological data were retrieved from prospectively maintained institutional databases. The following variables were extracted: gender, age, primary tumor origin, ASA score, ECOG Performance Status Scale [17], number of previous lines and cycles of systemic chemotherapy, presence of symptoms before PIPAC (abdominal pain, ascites, obstructive symptoms, nausea), Peritoneal Cancer Index (PCI) [18], Peritoneal Regression Grading Score (PRGS) [19], postoperative complications [16], and overall survival. Ascites volume was measured at PIPAC#1, based on the intraoperative aspirates. Bimodal treatment was defined as the patient being under systemic chemotherapy in the interval between PIPACs. Median PRGS was calculated from the scores of biopsies taken during each individual PIPAC.

### 2.3. PIPAC Procedure and Safety Considerations

The PIPAC procedure has been described previously and was applied according to current recommendations and safety protocols [9,20]. Oxaliplatin was applied at a dose of 92 mg/m^2^ for carcinomatosis of mostly colorectal origin and, selectively, for other digestive origins (gastric or pancreatic cancer). Cisplatin (7.5 mg/m^2^) in combination with doxorubicin (1.5 mg/m^2^), with dose adaptation (10.5 mg/m^2^ and 2.1 mg/m^2^) since 2019, was applied for the remaining cases [21].

### 2.4. Statistical Analysis

Continuous variables were presented as mean with standard deviation (SD) or median with interquartile range (IQR) according to their distribution. Categorical variables were reported as frequencies (%) and compared with chi-square test. Student’s *t*-test or Mann–Whitney test were used to compare continuous variables. Multivariable analyses were performed by using a multiple logistic regression integrating variables with univariate *p*-values ≤0.1. Kaplan–Meier survival curves were used to analyze time-to-event data and to compare two groups of subjects. All statistical tests were two-sided and a *p*-value of <0.05 was used to indicate statistical significance. Statistical analyses were performed with GraphPad Prism 8 (GraphPad Software, Inc., La Jolla, CA, USA).

## 3. Results

In total, 517 procedures were performed for 183 patients; 53 patients had only 1 PIPAC (29%), 35 had 2 PIPACs (19%), 60 had 3 PIPACs (33%) and 35 had >3 PIPACs (19%). Ninety-five patients (52%) had completed PP treatment. PM origin was ovarian in 59 (32%) patients, colorectal in 55 (30%), gastric in 37 (20%), HPB in 18 (10%), and mesothelioma in 14 (8%). No difference was observed between the two groups regarding PM origin (*p* = 0.52). Overall median survival was longer in the PP group compared to the <3 PIPAC group (16 vs. 7.2 months, *p* = <0.001).

The main reason for interrupting PIPAC treatment was oncological progression, in 44 (50%) patients (42% intraperitoneal and 8% extraperitoneal). All reasons for PIPAC interruption are described in Figure 1.

The absence of prior history of bowel obstruction before PIPAC1, lower volume of ascites (<500 mL) retrieved during the first procedure (PIPAC1), and bimodal treatment were associated with completion of PIPAC treatment (*p* = 0.028, *p* = 0.001, and *p* < 0.001, respectively) (Table 1). After multiple regression, bimodal treatment was found as an independent predictive factor for completing PP treatment (OR = 4.202, 95%CI [1.813, 10.630], *p* < 0.001), along with prior bowel obstruction (OR = 0.389, 95%CI [0.153, 0.920], *p* = 0.037).

Prior history of bowel obstruction and bimodal treatment were found as independent predictive factors of the discontinuation of PIPAC after multivariable logistic regression (OR = 0.389, 95%CI [0.153, 0.920], *p* = 0.037) and (OR = 4.202, 95%CI [1.813, 10.630], *p* < 0.001), respectively (Table 2).

Median PRGS at baseline was comparable in the two groups at PIPAC#1, 2 (IQR: 1.2–3.2) for the PP group and 2 (IQR: 2–2) for the <3 PIPACs group (*p* = 0.145). The mean PRGS score was lower in the PP group at PIPAC#3 compared to patients in the <3 PIPACs group at PIPAC#2, with 1 (IQR: 1–1.25) vs. 2 (IQR: 1–3), respectively (*p* = 0.009).

## 4. Discussion

The absence of ascites and prior bowel obstruction increases the chances of completing the three PIPAC procedures and best results seem to be achieved when PIPAC is combined with systemic chemotherapy. The association between better overall survival and complete PIPAC treatment is encouraging, but this finding should be taken with caution due to possible selection bias. Optimal patient selection appears mandatory, and caution is warranted in patients with obstructive symptoms, abundant ascites, and limited life expectancy.

This study confirmed the feasibility and safety of PIPAC treatment [6]. In fact, less than 3% of patients had to stop their treatment directly because of a technical problem during PIPAC (non-access, *n* = 4/183) or directly caused by the procedure (postoperative complications, *n* = 1/88). Of note, 76 (42%) patients in our series had prior major surgery (including 11 cytoreductive surgery + HIPEC). Although our study is modest in numbers, this point is important to note and can support the evaluation of PIPAC as a neoadjuvant or an adjuvant prophylactic treatment [22,23,24].

The early discontinuation of the planned protocol is a regularly reported problem for PIPAC as many retrospective series have a median number of 2.5 PIPAC/patient [25].

Oncologic progression remains the main reason for interruption of PIPAC treatment in half of the patients; this can be partially explained by a negative selection bias. Indeed, in the majority of published series [6,7], most patients receiving PIPAC are in palliative situations with advanced, aggressive, and refractory disease. However, we were not able to highlight a link between the oncological disease history and PIPAC treatment discontinuation. Neither PM origin, intraoperative tumor load (PCI), nor the amount of previous systemic chemotherapy (lines or cycles) seemed to play a role. Our findings suggest that an ascites-producing peritoneal disease might be possibly linked to early termination of PIPAC therapy. The presence of malignant ascites has already been reported to be associated with poorer prognosis [26]. Intraperitoneal and extraperitoneal progression underlined in our series supports the rigorous monitoring of the disease progression during the exploration phase of the PIPAC procedure, along with current recommendations [9,27]. In line with our results, the use of bimodal treatment, combining PIPAC with systemic chemotherapy might be a good option to limit tumoral progression. While bimodal or bidirectional treatment is regularly practiced and reported as feasible and safe [28,29], no clear data on a survival benefit could be confirmed up to date. Further evidence arising from large registry data is expected in the future [30,31].

Our results did not reveal any difference regarding general condition (ASA score) and performance status (ECOG scale) in patients with PP treatment or early termination. Furthermore, the proportion of patients with an ECOG ≥2 was similar between the two groups. This can be explained by the fact that PIPAC is mostly well-tolerated, with no negative impact on quality of life, and is thus applicable repeatedly even in patients with a certain degree of functional impairment [10]. In contrast, severe functional impairment (ECOG 3–4) is still currently a relative contra-indication to PIPAC treatment [32].

Nine patients (10%) decided to stop the treatment prematurely. No clear reason could be identified. Fear of surgery or general anesthesia and refusal of repeated hospitalizations were mentioned, but not systematically. The proportion of patients willing to withdraw PIPAC therapy found in our study is comparable to studies evaluating the discontinuation of systemic chemotherapy (10–20%) [33]. In a palliative setting, the patient’s decision to withdraw chemotherapy, whatever its administration route, is multifactorial and encompasses broader psychologic, physiologic, and personal aspects that are beyond the scope of the present study. PIPAC is probably similarly tolerated to systemic chemotherapy.

No correlation was found between pathological response to systemic chemotherapy at baseline and treatment discontinuation. Baseline PRGS was comparable in the two groups at PIPAC#1. Patients who completed PP treatment showed a better pathological response with a significantly lower PRGS and longer OS. The same observations were made regarding PIPAC for PM of appendicular origin in recent reports [14].

The current study has some limitations; these are mainly related to its retrospective nature and limited patient number. Small differences between the comparative groups might have passed undetected due to type II error. Although the two groups were comparable for ASA score, ECOG, and baseline symptoms, heterogeneity in previously administrated treatments and disease presentation are further limiting factors. Therefore, it would be suitable to extend this study to a larger group of patients with more selective inclusion criteria.

## 5. Conclusions

In conclusion, the reasons for stopping are multifold. Benefiting from a bimodal treatment and the absence of prior bowel obstruction before PIPAC1 increases the chances of completing the three procedures. Treatment completion is associated with better prognosis, although at this stage a direct relationship between PIPAC and overall survival cannot be established. The first published results of cohort studies on overall survival are still pending.

There is a need for further investigations in order to allow proper patient selection with good care and precise criteria. The elaboration of a predictive score for complete PIPAC treatment is part of the ongoing international PIPAC cohort study and could allow better patient selection criteria in the future.

## Figures and Tables

**Figure 1 cancers-14-02557-f001:**
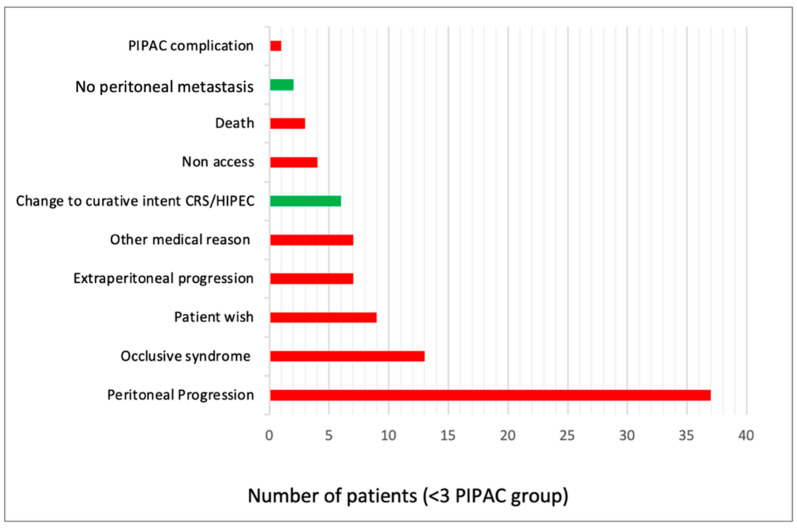
Reasons for PIPAC interruption before PIPAC#3. Data presented as number of patients. PIPAC—Pressurized IntraPeritoneal Aerosol Chemotherapy. CRS/HIPEC—Cytoreductive Surgery/Hyperthermic Intraperitoneal Chemotherapy.

**Table 1 cancers-14-02557-t001:** Potential predictors for completion of PIPAC treatment.

°	<3 PIPAC Procedures *n* = 88 Patients	≥3 Procedures (PP) *n* = 95 Patients	*p*-Value
Demographics			
Gender male, n (%)	28 (32)	38 (40)	0.632
Mean age (IQR)	65 (55–72)	63 (53–71)	0.638
ASA III–IV, n (%)	41 (47)	38 (40)	0.371
ECOG			0.189
0–1, n (%)	53 (60)	66 (70)	
≥2, n (%)	35 (40)	29 (30)	
Previous systemic chemotherapy			
≥3 lines, n (%)	29 (33)	35 (37)	0.644
≥12 cycles, n (%)	27 (31)	40 (42)	0.134
Symptoms before PIPAC procedures			
Abdominal pain, n (%)	22 (25)	19 (20)	0.472
Ascites, n (%)	24 (27)	18 (19)	0.219
Prior bowel obstruction (ileus), n (%)	21 (24)	11 (12)	0.028
Nausea, n (%)	14 (16)	7 (7)	0.150
Surgical details			
Median PCI (IQR) at PIPAC#1	19 (10–29)	18 (9–25)	0.213
Mean ascites (mL) (SD), at PIPAC#1	960 (188)	410 (100)	0.001
Intraperitoneal chemotherapy regimen			0.640
Oxaliplatin, n (%)	25 (28)	30 (32)	
Cisplatin + Doxorubicin, n (%)	63 (72)	65 (68)	
Bimodal treatment, n (%)	11 (13)	37 (39)	<0.001
Postoperative complications			
Overall, n (%)	31 (35)	42 (44)	0.210
Severe compilation III-IV, n (%)	7 (8)	5 (5)	0.462

Median (IQR—interquartile range or range), mean (SD—standard deviation), or number (%) as appropriate. Statistical significance (*p* < 0.05) is highlighted in bold. ASA: American Association of Anesthesiologists physical status classification system. ECOG Performance Status Scale [17]. PCI—Peritoneal Cancer Index [18].

**Table 2 cancers-14-02557-t002:** Multivariable logistic regression analysis correlating with completion of the per protocol treatment.

Baseline Variable	OR	95% CI for OR	*p*-Value
Prior bowel obstruction (yes vs. no)	0.389	0.153 to 0.920	0.037
Ascites ≥ 500 mL (yes vs. no)	0.649	0.304 to 1.35	0.254
Bimodal treatment (yes vs. no)	4.281	1.851 to 10.79	**0.001**

After univariate analysis, *p*-values ≤ 0.1 were incorporated in the multivariable analysis. Statistical significance (*p* < 0.05) is highlighted in bold. OR: odds ratio.

## Data Availability

Not applicable.

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
