# Peer review of "Selection Criteria for Pressurized Intraperitoneal Aerosol Chemotherapy (PIPAC) Treatment in Patients with Peritoneal Metastases"

_cancers, 2022, doi:10.3390/cancers14102557_

Round 1

Reviewer 1 Report

An interesting multicenter cohort study for PIPAC for patients with peritoneal metastases. Aim of the study was to identify predictors for completion of treatment, and the authors compared patients three or more PIPAC sessions versus patients completing less than 3 PIPAC sessions. Patients were intended to receive 3 PIPAC sessions, patients received less in case of tumor progression and logically overall survival was significantly better for patients receiving 3 or more PIPAC sessions. Some parameters were independent predictors, these included bimodal treatment -combination with systemic chemotherapy- , prior bowel obstruction and absence of ascites. The institutes analyzed a decent number of 183 patients treated in 2015-2020.

A nice and valuable analysis but the reader is left with the problem that any conclusion regarding a correlation between number of PIPAC sessions and treatment outcome is dubious as patients receives less PIPAC sessions if tumor progress was observed, and obviously early tumor progress can also indicate more aggressive disease. The other conclusions are better supported by the results of this retrospective analysis: bimodal treatment, absence of prior bowel obstruction and absence of ascites are all correlated with better overall survival.

When the authors state in the first line of the discussion that the results of this retrospective study suggest that a benefit from incomplete (i.e. less than 3 sessions) PIPAC treatment is unlikely, then I feel that they have not yet shown that that is a direct correlation, not even partly, so I feel that this is indirect causality and the apparent correlation can be completely or partially explained by a negative selection bias as the authors also mention. They should rephrase their conclusions to reflect that difference in evidence and add more caution in their conclusions.

The authors also refer to a larger study which yielded similar conclusions. When I checked reference 14 Farinha et al EJSO I found this is a conference abstract and that Lausanne was also part of that study, so possibly in part the same patients were analyzed? Please explain to what extent there is overlap between the study reported in ref 14 and the present study. And note that you messed up/obliterated the name of the second author Somashekhar in ref 14!

Author Response

A nice and valuable analysis but the reader is left with the problem that any conclusion regarding a correlation between number of PIPAC sessions and treatment outcome is dubious as patients receives less PIPAC sessions if tumor progress was observed, and obviously early tumor progress can also indicate more aggressive disease. The other conclusions are better supported by the results of this retrospective analysis: bimodal treatment, absence of prior bowel obstruction and absence of ascites are all correlated with better overall survival.

Thank you for this comment, we completely agree with this point. This study does not attempt to include patients with very aggressive disease who will of course progress more quickly and not have all 3 procedures. We cannot draw any conclusion on the influence of the number of PIPACs on the outcomes. The purpose of this study is to better select patients with a high probability of benefiting from a complete treatment. The text has been rephrased.

When the authors state in the first line of the discussion that the results of this retrospective study suggest that a benefit from incomplete (i.e. less than 3 sessions) PIPAC treatment is unlikely, then I feel that they have not yet shown that that is a direct correlation, not even partly, so I feel that this is indirect causality and the apparent correlation can be completely or partially explained by a negative selection bias as the authors also mention. They should rephrase their conclusions to reflect that difference in evidence and add more caution in their conclusions.

We totally agree with this point. We are still awaiting the first results of the cohort studies. One could extrapolate that if the planned treatment can be given in its entirety the chances of a better oncological response are greater. However currently there is no evidence. We will rephrase the conclusion more cautiously.

The authors also refer to a larger study which yielded similar conclusions. When I checked reference 14 Farinha et al EJSO I found this is a conference abstract and that Lausanne was also part of that study, so possibly in part the same patients were analyzed? Please explain to what extent there is overlap between the study reported in ref 14 and the present study. And note that you messed up/obliterated the name of the second author Somashekhar in ref 14!

Absolutely, some patients are involved in both cohorts. However, the overlap is minimal. The study relating to reference 14 only includes patients with peritoneal carcinomatosis of appendicular origin, which is not the case for the present study. Moreover, the aim of the cited study is to evaluate the oncological response to PIPAC and not to investigate predictive factors of PIPAC treatment discontinuation.

Thank you for pointing out this unfortunate mistake in the references. We corrected it. (reference 14 & 15)

Reviewer 2 Report

The paper is well written and of high interest in a new filed of oncological treatments. The selection criteria for PIPAC treatment are one of the most important Gordian knots to obtain positive results and decrease complications.

Results and discussion are exhaustive such as referencies.

I strongly reccomend the pubblication in present form

Author Response

We thank reviewer 2 for their encouraging comments.

Round 2

Reviewer 1 Report

I thank the authors for addressing my earlier concerns